



# Methane oxidation in the waters of a humics-rich boreal lake stimulated by photosynthesis, nitrite, Fe(III) and humics

Sigrid van Grinsven[1*], Kirsten Oswald[1,2*], Bernhard Wehrli[1,2], Corinne Jegge[1,3], Jakob Zopfi[4], Moritz F. Lehmann[4] & Carsten J. Schubert[1,2]

[1]Department of Surface Waters – Research and Management, EAWAG, Swiss Federal Institute of Aquatic Science and Technology, Kastanienbaum, Switzerland

[2]Institute of Biogeochemistry and Pollutant Dynamics, ETH Zurich, Swiss Federal Institute of Technology, Zurich, Switzerland

[3]School of Architecture, Civil and Environmental Engineering, EPFL, Swiss Federal Institute of Technology, Lausanne, Switzerland

[4]Department of Environmental Sciences, Aquatic and Stable Isotope Biogeochemistry, University of Basel, Basel, Switzerland

*Correspondence to*: sigrid.vangrinsven@eawag.ch

*These authors contributed equally to this work.

Running title: Methane oxidation in Lake Lovojärvi

Key words: anaerobic methane oxidation, anthraquinonedisulfonate, nitrite, AQDS, photosynthesis, ferrihydrite, manganese oxide, Lovojärvi



## 1 Abstract

Small boreal lakes are known to contribute significantly to global methane emissions. Lake Lovojärvi
is a eutrophic lake in Southern Finland with bottom water methane concentrations up to 2 mM. However,
the surface water concentration, and thus the diffusive emission potential, was low (<0.5 µM). We
studied the biogeochemical processes involved in methane removal by chemical profiling and through
incubation experiments. $\delta^{13}$C-CH$_4$ profiling of the water column revealed methane-oxidation hotspots
just below the oxycline and within the anoxic water column. In incubation experiments involving the
addition of light and/or oxygen, methane oxidation rates in the anoxic hypolimnion were enhanced 3-
fold, suggesting a major role for photosynthetically fueled aerobic methane oxidation. A distinct peak
in methane concentration was observed at the chlorophyll a maximum, caused by either in-situ methane
production or other methane inputs such as lateral transport from the littoral zone. In the dark anoxic
water column at 7 m depth, nitrite seemed to be the key electron acceptor involved in methane oxidation,
yet additions of Fe(III), anthraquinone-2,6-disulfonate and humic substances also stimulated anoxic
methane oxidation. Surprisingly, nitrite seemed to inhibit methane oxidation at all other depths. Overall,
this study shows that photosynthetically fueled methane oxidation can be a key process in methane
removal in the water column of humic, turbid lakes, thereby limiting diffusive methane emissions from
boreal lakes. Yet, it also highlights the potential importance of a whole suite of alternative electron
acceptors, including humics, in these freshwater environments in the absence of light and oxygen.

## 20 Introduction

Lacustrine water bodies represent a substantial natural source of atmospheric methane (CH$_4$), a major
contributor to global warming. They may release up to ~72 Tg CH$_4$ a$^{-1}$ (12 % of total global emissions)
(Bastviken et al., 2011), despite covering a relatively small proportion of the land surface area (>3%,
Downing et al. 2006). In temperate and northern boreal regions, small lakes generally emit more CH$_4$
per unit area than larger systems (Juutinen et al., 2009; Kortelainen et al., 2000, 2004; Michmerhuizen
et al., 1996). Northern lakes alone are estimated to contribute 24.2±10.5 Tg CH$_4$ a$^{-1}$ to global methane
emissions (Walter et al., 2007).
The majority of lacustrine methane is produced by anaerobic methanogenic archaea as the end product
of remineralization of organic matter in anoxic sediments (Bartlett and Harriss, 1993; Rudd, 1980). From
the sediments, methane can diffuse into the water column and may be emitted to the atmosphere at the
water-air interface. Physical factors including stratification regime, mixing events, vertical diffusion and
bubble formation affect how much methane reaches the upper water layers (Bastviken et al., 2004;
Lehmann et al., 2015; McGinnis et al., 2006; Michmerhuizen et al., 1996; Riera et al., 1999). Large
fractions of the methane that is produced by methanogenesis in sediments and anoxic parts of lacustrine



water columns may be consumed by microbial methane oxidation, decreasing the methane concentration
and thus limiting methane emissions. Research has shown that microbial methane oxidation may be the
single most important control on methane emissions from lakes and other ecosystems (Chistoserdova,
2015), thus also referring to methane oxidizing microbes as the "biological methane filter".
The vast majority of $CH_4$ consumption in limnic systems has been assigned to aerobic methane oxidation
(Hanson and Hanson, 1996; King, 1992). This process is performed by methane-oxidizing bacteria
(MOB), affiliated with either gamma- (type I and type X) or alphaproteobacteria (type II). Although the
majority of MOB are unicellular, it has been shown recently that filamentous gamma-MOB related to
*Crenothrix polyspora* also actively turn over methane in freshwater lakes (Oswald et al., 2017). For all
types of aerobic methanotrophs, methane oxidation is a multi-step enzymatic pathway mediated by
particulate- (pMMO) or soluble methane monooxygenase (sMMO) in the first oxidative step from $CH_4$
to methanol (Hanson and Hanson, 1996). As most MOB express the functional gene encoding for
pMMO (*pmoA*), it is commonly used for environmental detection of these organisms (Knief, 2015).
Typically, oxygen is required for $CH_4$ activation with pMMO and also as the terminal electron acceptor
(TEA) in the respiratory chain. However, some aerobic gamma-MOB like *Methylomonas denitrificans*
(Kits et al., 2015a) and *Methylomicrobium album* (Kits et al., 2015b) can switch to using nitrate ($NO_3^-$)
or nitrite ($NO_2^-$) as their TEA, respectively, even at trace-amount levels of $O_2$ (<50 nM) that still maintain
a functioning pMMO. Similarly, the hybrid metabolism of *Methylomirabilis oxyfera* combines partial
denitrification ($NO_2^-$ to NO) and classical MO, fueled by internal $O_2$ generation (splitting NO to $N_2$ and
$O_2$) (Ettwig et al., 2010). While *M. oxyfera* has similar metabolic traits as proteobacterial methanotrophs,
it is associated with the novel phylum NC10 (Holmes et al., 2001; Rappé and Giovannoni, 2003).
Completetely $O_2$-independent methane consumption by anaerobic oxidation of methane (AOM) is
assigned to three distinct groups of anaerobic methanotrophic archaea (ANME-1, -2 and -3), which, at
least in marine settings, are often found in syntrophic relationship with sulfate-reducing bacteria
(Boetius et al., 2000; Michaelis et al., 2002; Orphan et al., 2001). In ocean sediments and water columns
AOM mediated by ANME  accounts for >90% of the oxidized $CH_4$ (Hinrichs and Boetius, 2002;
Reeburgh, 2007). Although rare, ANME can be present in lake waters (Durisch-Kaiser et al., 2011; Eller
et al., 2005; Oswald et al., 2016a) and sediments (Schubert et al., 2011; Su et al., 2020). Interestingly,
studies reporting on methane oxidation in anoxic zones of lakes in the absence of ANME and in the
presence of aerobic MOB are increasing (Biderre-Petit et al., 2011; Blees et al., 2014; van Grinsven et
al., 2020b; Oswald et al., 2016b; Schubert et al., 2010). While oxygen supplied by episodic down-
welling of cold water (Blees et al., 2014), or low-light photosynthesis (Milucka et al., 2015; Oswald et
al., 2015) may explain this phenomenon to some degree, electron acceptors such as $NO_X$ (Deutzmann
et al., 2014; Graf et al., 2018; Oswald et al., 2016b), Fe(III) (Norði et al., 2013; Sivan et al., 2011),
Mn(IV) (Crowe et al., 2011; Oswald et al., 2016a) and humic substances (Valenzuela et al., 2019) can
are likely to play, to some extent, a role as well.




Given the widespread distribution of boreal lakes and their contribution to global methane emissions,
studies focusing specifically on methane oxidation (as well as the microorganisms involved) in such
systems are relatively scant (Kankaala et al., 2007; Sundh et al., 2005). Moreover, the environmental
controls on the modes of AOM in these lakes, and the TEAs involved, are still poorly understood. Here,
we studied the microbial methane turnover, in particular the oxidative side, in a small humic-substances-
rich lake in southern Finland (Lake Lovojärvi). Sedimentation regime, stratigraphy and phytoplankton
community have been studied intensively in this lake (Keskitalo, 1977; Saarnisto et al., 1977; Simola et
al., 1990). However, only little is known about its carbon and methane dynamics (Mutyaba, 2012), let
alone the corresponding microbial aspects. To shed light on the fate of biogenic methane in Lake
Lovojärvi, and to gain a more mechanistic understanding on the microbial and biogeochemical controls
on its biological methane filter, we combined physical and chemical water column profiling, incubation
experiments with different TEAa to quantify methane turnover rates and modes, as well as molecular
techniques to characterize the key microbial players involved.

## Materials and Methods

### Study site

Lake Lovojärvi is a small (5.4 ha) eutrophic lake near the town of Lammi in southern Finland. It is part
of a glaciofluvial esker deposit (Simola, 1979), which gives the lake its elongated shape (600 m long,
130 m wide) and shields it from strong winds (Hakala, 2004). Lake Lovojärvi is shallow, with an average
depth of 7.7 m (Ilmavirta et al., 1974) and a maximum depth of 17.5 m in the southeastern part (Simola,
1979). Due to the sheltered location and basin morphology, the lake undergoes strong thermal
stratification and has a permanently anoxic hypolimnion (Saarnisto et al., 1977). The catchment of Lake
Lovojärvi is 7.2 km$^2$ and drains water from predominantly agricultural and swampy areas (Simola, 1979).
Hydrologically connected to marsh/wetlands (Limminjärvi), the lake receives high inputs of humic
substances and dissolved ions (Hakala, 2004). To our knowledge, no information on groundwater inflow
is available.

### In situ profiling and sample collection

Profiling and sample collection were carried out in September 2015, at the deepest part of the lake (61°
04.584'N, 25°02.116'E). A custom-made profiling device equipped with various probes and sensors
was used to measure the following parameters in situ: conductivity, turbidity, temperature, depth
(pressure) and pH (XRX 620, RBR); photosynthetically active radiation (PAR; LI-193 Spherical
Underwater Quantum Sensor, LI-COR); chlorophyll a (ECO-FL, Wetlands, EX/EM= 470/695); and





dissolved $O_2$ (micro-optodes PSt1 and TOS7, PreSens). The detection limits of the two $O_2$ optodes were
125 and 20 nM, respectively.
Samples for the analysis of all other parameters were pumped to the surface with a peristatic pump
(Zimmermann AG Elektromaschinen, Horw, Switzerland) connected to gas tight tubing (PVC Solaflex,
Maagtechnic) attached to the profiler. To guarantee that water was taken from the correct depth, a
custom-built inlet system was used (designed after Miracle et al., 1992) and water was pumped for 2
minutes (time necessary to replace the entire tube volume) prior to filling 60 mL syringes directly from
the tube outlet avoiding air contact. Water from the syringes was then sub-sampled into different vials
for further processing: For total sulfide analysis ($HS^-+H_2S$) zinc acetate was added (1.3% final
concentration). To quantify dissolved (<0.45 µm) and total fractions of metals, iron(II)/(III) and organic
carbon, samples were acidified immediately to a final concentration of 0.1 M (Suprapur $HNO_3$, Merck),
0.5 M (HCl) and 0.02 M (HCl), respectively. Aliquots were sterile filtered (<0.22 µm) to analyze
concentrations of dissolved nitrogen species ($NO_3^-$, $NO_2^-$ and $NH_4^+$), sulfate ($SO_4^{2-}$), phosphate ($PO_4^{3-}$)
and dissolved inorganic carbon (DIC). DIC samples were filled into gas-tight 12 mL Exetainers (Labco
Ltd) without a headspace, and stored upside down. Water samples intended for hybridization techniques
was fixed immediately with formaldehyde (2 % [v/v] final concentration), and stored in the dark at 4°C.
All other samples requiring larger water volumes were taken directly from the tube outlet anoxically
(without headspace or bubbles and by letting water overflow 2-3 volumes). For methane concentration
and isotopic measurements, 120 mL serum bottles were filled prior to adding Cu(I)Cl (~0.15 % [w/v]
final concentration) and sealing the bottles with butyl stoppers (Geo-Microbial Technologies, Inc.) and
aluminum crimp caps. Similarly, sterile 160 mL serum bottles or 1 L Schott bottles served to store water
for incubation experiments and DNA analysis. These were sealed with butyl stoppers and crimp or screw
caps, and were kept in the dark at 4 °C.
**Carbon and isotopic parameters**
After generating a 20 mL $N_2$ headspace and equilibration, dissolved $CH_4$ concentrations were measured
by gas chromatography (GC; Agilent 6890N, Agilent Technologies) using a Carboxen 1010 column (30
m x 0.53 mm, Supelco), a flame ionization detector and an auto-sampler (Valco Instruments Co. Inc.).
Resulting headspace concentrations were converted to dissolved water-phase $CH_4$ by applying
calculated Bunsen solubility coefficients (Wiesenburg and Guinasso, 1979). Stable carbon isotopes of
$CH_4$ were analyzed in the same headspace by isotope ratio mass spectrometry (IRMS; GV Instruments,
Isoprime). For this, injected gas samples first passed through a trace gas unit (T/GAS PRECON,
Micromass UK Ldt) for purification, concentration, and combustion to $CO_2$ (for details see Oswald et
al., 2016a, 2016b). Isotopic ratios of $^{13}C/^{12}C$ are presented in the standard $\delta^{13}C$-notation (relative to the
Vienna Pee Dee Belemnite (VPDB) reference) with a precision of ~1.2 ‰.





Based on the methane concentration profile and the corresponding isotopic ratios, fractionation factors
for methane oxidation ($\alpha_c$) were determined with the Rayleigh Equation (Whiticar and Faber, 1986):

$$\delta^{13}C = \left[ \delta^{13}C_0 + 1000 \cdot f^{\left(\frac{1}{\alpha_c}-1\right)} \right] - 1000$$

$\delta^{13}C$ and $\delta^{13}C_0$ represent the $^{13}C/^{12}C$ isotopic ratios of $CH_4$ at the top and at the bottom of the oxidation
zone, respectively. The fraction of remaining methane above this same zone is denoted with $f$
(calculated as the ratio of the $CH_4$ concentration at a given depth and the concentration at the bottom of
the oxidation zone).
Total organic carbon (TOC), dissolved organic carbon (DOC) and DIC were quantified with a total
carbon analyzer (TOC-L, Schimadzu) equipped with a nondispersive infrared detector (NDIR). TOC
was measured as $CO_2$ after combustion (680 °C) of the untreated sample. For DOC determination, the
samples were acidified before combustion. For DIC analysis, unacidified samples were injected and
DIC was volatilized to $CO_2$ (internal addition of HCl, pH <3, in a $CO_2$-free closed reaction chamber)
and quantified subsequently. For carbon isotope analysis, 1 mL of the remaining liquid was then
transferred to a He-flushed 3.7 mL exetainer and acidified (100 µl 85 % $H_3PO_4$). The $\delta^{13}C$-DIC of the
released $CO_2$ (overnight equilibration) was measured with a gas-bench system (MultiFlow, Isoprime)
connected to an IRMS (Micromass, Isoprime). Isotopic ratios of the DIC are also expressed in the $\delta^{13}C$-
notation (VPDB reference) with a precision of ~0.15 ‰.

### Nutrients and metals

Nitrite, ammonium, sulfide and iron(II)/(III) concentrations were measured on the same day as sampled
using photometric protocols according to Griess (1879), Krom (1980), Cline (1969) and Stookey (1970),
respectively. Fe(III) concentrations were determined as the difference between total iron, after reduction
with hydroxylamine hydrochloride, and Fe(II), which was measured directly (Viollier et al., 2000).
Concentrations of nitrate and phosphate were quantified by flow injection analysis (SAN++, Skalar),
and sulfate concentrations were determined by ion chromatography (882 Compact IC plus, Metrohm).
Total and dissolved Mn concentrations were analyzed by inductively coupled plasma-mass spectrometry
(ICP-MS; Element2, Thermo-Fisher).

### Catalyzed reporter deposition – fluorescence in situ hybridization (CARD-FISH)

Formaldehyde-fixed samples (incubated for ~12 h at 4 °C) were filtered onto 0.2 µM polycarbonate
filters (GTTP, Millipore) and rinsed 2x with 1x phosphate buffered saline. Filters were stored at -20 °C
until standard CARD-FISH (Pernthaler et al., 2002) was carried out using specific oligonucleotide
probes with horseradish peroxidase labels (purchased from Biomers) An overview of the primers and
percentage formamide used is supplied in Table S1. Probes EUB338 I-III and Mgamma84+705 were



applied as a mix of equal proportions. Background signals were assessed with probe NON338.
Permabilization of cell walls, inactivation of endogenous peroxidase activity, hybridization,
amplification (Oregon Green 488, Thermo-Fischer Scientific), counter staining (4',6-diamidino-2-
phenylindole, DAPI) and embedding of the filter pieces was carried out as described in detail previously
(Oswald et al., 2016b). Total cell numbers (DAPI-stained cells) and cells belonging to the different
targeted groups (CARD-FISH signals) were enumerated in 20 randomly selected fields of view using
the grid ocular of the Axioskop 2 (Zeiss) epifluorescence microscope. Proportions of the microbial
groups are based on total DAPI cell counts.

### DNA extraction and 16S rRNA gene amplicon sequencing

Microbial biomass from different depths of the water column was collected on 0.2 μm polycarbonate
membrane filters (Cyclopore, Whatman) and kept frozen (-20 °C) until DNA extraction using the
FastDNA SPIN Kit for Soil (MP Biomedicals). A two-step PCR approach (Monchamp et al., 2016) was
applied in order to prepare the library for Illumina sequencing at the Genomics Facility Basel. Briefly,
10 ng of extracted DNA were used, and a first PCR of 25 cycles was performed using universal primers
515F-Y (5′-GTGYCAGCMGCCGCGGTAA) and 926R (5′-CCGYCAATTYMTTTRAGTTT-3')
targeting the V4 and V5 regions of the 16S rRNA gene (Parada et al., 2016). The primers of this first
PCR were composed of the target region and an Illumina Nextera XT specific adapter sequence. Four
sets of forward and reverse primers, which contained 0-3 additional and ambiguous bases after adapter
sequence, were used in order to introduce frame shifts to increase complexity (details described in Su et
al, bioarxiv, 2021). Sample indices and Illumina adaptors were added in a second PCR of 8 cycles.
Purified, indexed amplicons were finally pooled at equimolar concentration, denatured, spiked with 10 %
PhiX, and sequenced on an Illumina MiSeq platform using the 2×300 bp paired-end protocol (V3-Kit).
The initial sequence treatment was done at the Genetic Diversity Center (ETHZ) where FastQC (v 1.2.11;
Babraham Bioinformatics) was used to check the quality of the raw reads and FLASH (Magoč and
Salzberg, 2011) to merge forward and reverse reads into amplicons of about 374 bp length, allowing a
minimum overlap of 15 nucleotides and a mismatch density of 0.25. Full-length primer regions were
trimmed using USEARCH (v10.0.240), allowing a maximum of one mismatch. Merged and primer-
trimmed amplicons were quality-filtered (size range: 250-550, no ambiguous nucleotides, minimum
average quality score of 20) using PRINSEQ (Schmieder and Edwards, 2011). OTU (operational
taxonomic unit) clustering with a 97 % identity threshold was performed using the UPARSE-OTU
algorithm in USEARCH v10.0.240 (Edgar, 2010, 2013). Taxonomic assignment of OTUs was done
using SINTAX (Edgar, 2016) and the SILVA 16S rRNA reference database v128 (Quast et al., 2013).
Downstream sequence analyses were done in R v3.5.1 using Phyloseq v1.25.2 (McMurdie and Holmes,
2013). The 16S rRNA amplicon reads (raw data) have been deposited in the NCBI Sequence Read
Archive (SRA) under BioProject number XXXXXX (will be provided before publication).




**Methane oxidation incubation experiments**
To determine the methane oxidation potential and possible stimulation by potential electron acceptors,
incubation experiments were setup with water from 3, 4, 5, 7 and 9 m depth no later than 2 h after
sampling. The approach is described in detail by Oswald et al. (2016b), adapting procedures described
for 15N incubations in Holtappels et al., (2011) . Briefly, water collected in 160 mL serum bottles was
first degassed (10-15 min with He) and then individually amended with the different electron acceptors
tested, except for the dark and light setups (Table S2). After this, 5 mL of a saturated $^{13}CH_4$ (99 atom%,
Campro Scientific) solution was injected under anoxic and sterile conditions into each bottle to a final
concentration of ~50 µM $CH_4$. Finally, water was dispensed into 12 mL exetainers without headspace,
and incubated at ~8°C (average lake temperature between 3-9 m) under dark or light (~5 µE m$^{-2}$ s$^{-1}$)
conditions. At selected time points (~0, 6, 12, 24 and 48 h), $ZnCl_2$ (200 µl, 50 % [w/v] solution) was
used to stop microbial activity in one exetainer per setup to analyze $\delta^{13}C$-DIC by GC-IRMS (see above).
Methane oxidation rates were estimated by linear regression of the change of $^{13}C$-DIC over the
experimental interval, under consideration of the in situ DIC concentration at the different incubation
depths (1-1.2 mM) (for details see Oswald et al., 2015, 2016a). For comparison between all setups and
depths, the MO potential was determined always over the initial 24 h time interval, when the production
of $^{13}C$-DIC was always linear.

**Results**
**Physicochemical conditions in the water column**
Oxygen concentrations were around 250 µM in the top 2 m of the Lake Lovojärvi water column (Fig.
1A). Below, the $O_2$ profile displayed a sharp gradient between 2-3 m depth, and complete oxygen
depletion was observed already below 3.1 m. A small peak in the $O_2$ concentration was observed
between 3 and 3.1 m depth (Fig. S1). The thermo- and pycnoclines were evidenced by gradients in
temperature between 3-5 m (surface temperature 13 °C, bottom 5 °C) and in salinity between 12-14 m,
respectively (Fig. 1A). Compared to the total radiation at the surface, PAR decreased from 27% (80 µE
m$^{-2}$ s$^{-1}$) at 0.6 m to 1% (3 µE m$^{-2}$ s$^{-1}$) at 2.2 m (Fig. 2). Light was still detected down to 6.6 m (0.01 µE
m$^{-2}$ s$^{-1}$; Fig. 2). Nitrate concentrations peaked between 4-7 m, with the highest concentrations of 19 µM
at 5.25 m (Fig. 1C). Above and below the nitrate peak, $NO_3^-$ concentrations averaged at 0.3 µM. A nitrite
peak was visible at similar depths, but with the maximum concentration found at 7 m (3 µM, Fig. 1C).
Below 12 m, $NO_2^-$ increased to 4 µM (Fig. 1C). Sulfate concentrations in the top were relatively invariant
around 150 µM, and declined sharply to ~12 µM at 12 m depth, whereas total sulfide was <1 µM down
to 9 m, from where it increased steadily to ~14 µM at 14 m (Fig. S2). Fe(III) showed a peak at 4–9 m
depth, with a maximum of 23 µM at 8 m (Fig. 1E). Dissolved Fe(II) increased from 8 m downwards to



reach a concentration of 830 µM at 17 m (Fig. S2). Manganese concentrations were much lower than
those of iron, with particulate Mn(IV) ranging around 0.3 µM showing subtle peaks at 4.5 m (0.7 µM)
and 11 m (1.7 µM; Fig. 1E). Dissolved Mn(II) was nearly undetectable in the top 3 m of the water
column (100 nM average), yet reached rather constant values of ~2 µM below (3-11 m), before
increasing towards the sediment (16 µM at 17 m, Fig. S2).
**Fig. 1.** Physicochemical characteristics, methane oxidation rates (MOR; under ambient conditions or
upon addition of potential inorganic/organic electron acceptors) and methane oxidizing bacterial
(MOB) abundance in the Lake Lovojärvi water column in September 2015. POC – Particulate organic
carbon. DOC – Dissolved organic carbon. Note the break at the [CH₄] axis of panel B and the axes of
MOR (MOR axes apply to both the upper (B, C) and lower panels (E and F). Methane oxidation rates
and error margins are also provided in Table S3.

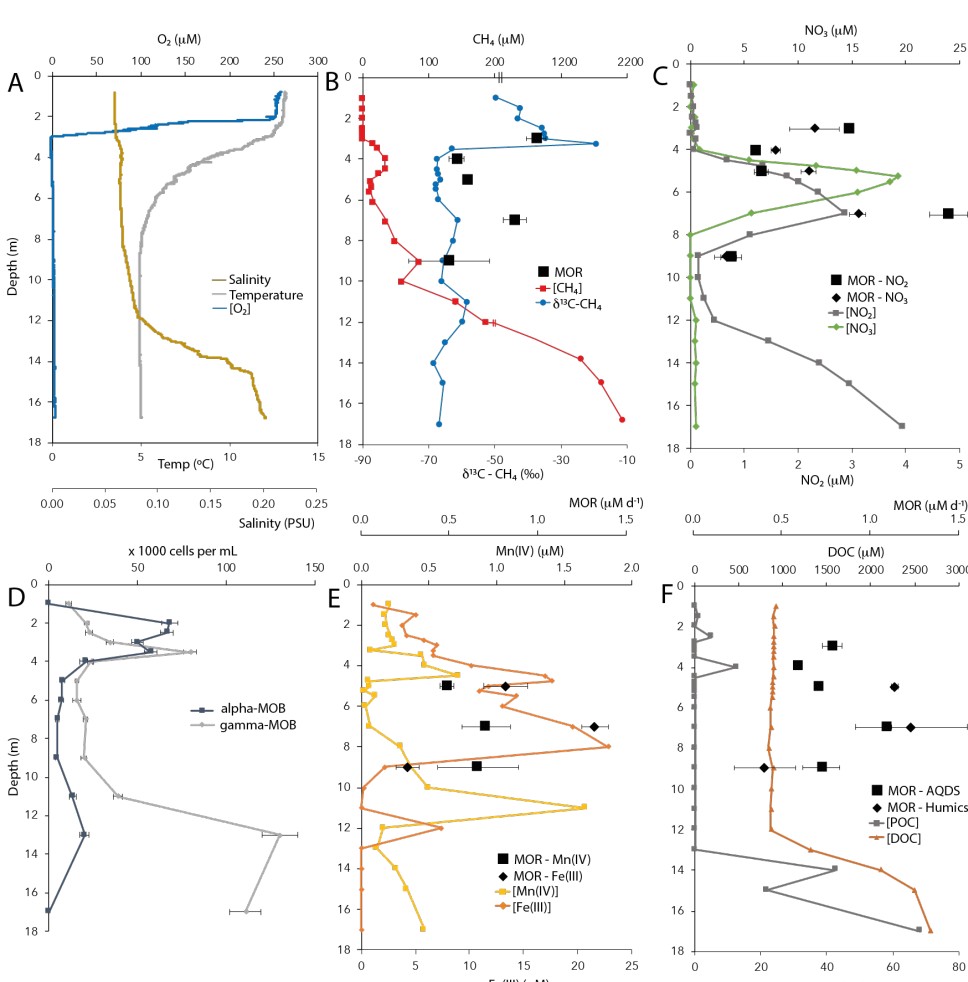







**Fig. 2.** Chlorophyll, light intensity (PAR) and dissolved oxygen in the water column of Lake
Lovojärvi along with the methane oxidation rates (MOR) measured in the dark (control), light and
oxygen-addition incubations. Note the different scale for MOR compared to Fig. 1.

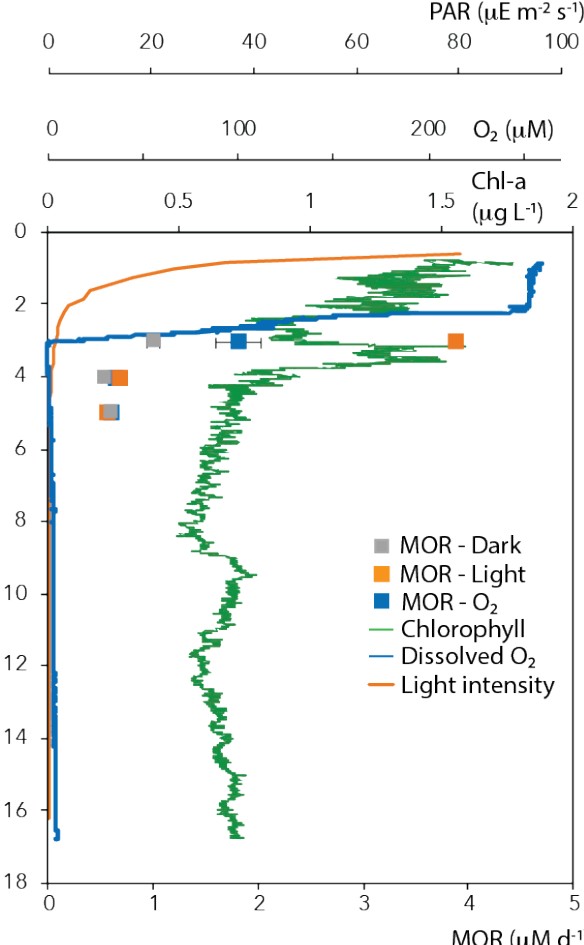




## Methane and carbon compounds

Methane was present throughout the water column of Lake Lovojärvi, yet increased by more than four orders of magnitude from the surface (0.3 µM) to the sediment (~2 mM; Fig. 1B). The profile exposed four 'zones': i) Low (≤0.3 µM) concentrations in the epilimnion, ii) a distinct peak in [$CH_4$] below the oxycline, from 3-5 m (max concentration 33 µM), iii) a zone of gradual increase, from 11 µM at 5.5 m to 140 µM at 11 m, and iv) a zone of rapid increase, from 190 µM at 12 m to 1990 µM at 17 m (Fig. 1B). The $\delta^{13}C$-$CH_4$ profile showed values of -50 ‰ to -35 ‰ in the epilimnion and of -58 to -69 ‰ in the hypolimnion, with a trend towards heavier values directly at the oxycline: the $\delta^{13}C$-$CH_4$ increased from -63 ‰ (3.5 m) to -19 ‰ (3.25 m), to decline to -35 ‰ at 3 m (Fig. 1B).

The majority of organic carbon was present in its dissolved form, with DOC concentrations being 100x higher than POC concentrations (Fig. 1F). Both DOC and POC profiles showed a constant concentration from the surface to the chemocline at 12 m depth, where both DOC and POC concentration profiles indicated a strong increase towards the sediment surface.

The DIC concentration profile followed that of $CH_4$ closely. Concentrations of DIC also increased by an order of magnitude from the surface (700 µM) to the sediment (5.6 mM), with a peak just below the oxycline (Fig. S3). $\delta^{13}C$-DIC values decreased from the surface waters (-11.5 ‰) to the oxycline (-18 ‰), remained relatively constant to 12 m depth, and then increased strongly towards the sediment (-4 ‰ at 17m; Fig. S3), a trend that could not be linked to that of $\delta^{13}C$-$CH_4$ (Fig. 1B).

## Microbial community and chlorophyll a distribution

Cell counts showed that both gamma- (probes Mgamma84+705) and alpha-MOB (probe Ma450) abundances showed a distinct peak near the oxycline (Fig. 1D). Gamma-MOB were present at all sampled depths, with peaks at 3.5 m ($8.0 \cdot 10^4$ cells $mL^{-1}$; 1.8% of DAPI counts), and in the hypolimnion at 13 m ($1.3 \cdot 10^5$ cells $mL^{-1}$; 3.5% of DAPI counts). Alpha-MOB were most numerous near the oxycline at 2 – 3.5 m, where they comprised a relatively large proportion of the total community ($6.8 \cdot 10^4$ cells $mL^{-1}$; 3.6 % of DAPI counts). A second, smaller peak was observed at 13 m ($2.0 \cdot 10^4$ cells $mL^{-1}$, 0.5 % of DAPI counts). Both types of MOB were least abundant between 4-9 m depth. Known representatives of ANME-1 (probe ANME-1-350) and ANME-2 (probe ANME-2-538) did not exceed 0.4 % of total DAPI counts at any depth of the water column (data not shown).

16S rRNA gene sequencing data showed that the archaeal relative abundance was below 0.5 % throughout the upper- and middle water column. Only between 11 and 17 m depth, the archaeal abundance was higher than 0.5 % (0.7, 1.0 and 4.0 % of all reads at 11, 13 and 17 m, respectively). The only known archaeal methanogens present belonged to the genus *Methanoregula* and were detected at 9, 11 and 17 m depth (0.1, 0.1 and 0.3 %; at all other depths <0.05 % and thus considered insignificant).



Gammaproteobacterial methane-oxidizing bacteria reads were detected throughout the water column,
and were dominantly assigned to the genus *Methylobacter* (0.3 – 5 % of total 16S rRNA reads) and to a
lesser extent to the genus *Crenothrix* (0 – 0.3 %; Fig 3). *Methyloparacoccus* dominated the oxic
epilimnion (0.9 – 1.1 %; Fig. 3), but was undetectable below 3.5 m depth. At 3.5, 13 and 17 m,
respectively 0.3, 0.1 and 0.3 % of 'other Methylococcaceae' were found. Alphaproteobacteria were
highly abundant in the oxic water column (14 – 15 %), but only 0.1 – 0.3 % of these reads was assigned
to the genus *Methylocystaceae*. 30 – 35 % of the Alphaproteobacterial reads at 2 – 3 m depth was,
however, assigned to unknown bacteria of the Rhizobiales order, to which *Methylocystaceae* belong
(Fig. S4). Bacteria of the family Methylophilaceae were present throughout the water column (0.6 –
2.3 %, Fig. 3). Sequence reads of *Canditatus* Methylomirabilis sp. were detected only at one single depth
(13 m) but at a comparatively high relative abundance (2.3 %). The genus *Acidovorax* was highly
abundant (19 – 40 % of total reads at 3.5 – 13 m depth) in the anoxic water column, except at 17 m
(5 %), whereas Planctomycetaceae were specifically abundant in the oxic water column (6 – 17 % at 2
– 3 m depth).
Chlorophyll a was present throughout the water column (Fig. 2). Yet, concentrations were highest in the
surface waters (1.8 µg L$^{-1}$), from where they decreased towards 2 m depth. A second peak in chlorophyll
a was visible at 3-4 m depth (1.6 µg L$^{-1}$; Fig. 2).





**Fig. 3.** Relative abundance of 16S rRNA gene sequences annotated to the methanotrophic genera
*Methylobacter*, *Methyloparacoccus* and *Crenothrix*, and the methylotrophic family Methylophilaceae
in the water column of Lake Lovojärvi.

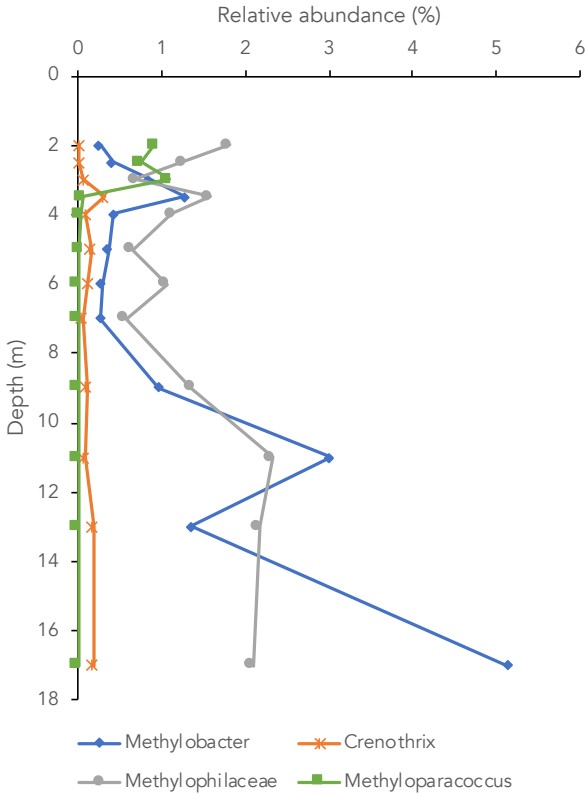


## Methane oxidation rate incubations

Methane oxidation under mimicked natural conditions (dark, starting concentration ~50 μM $CH_4$ after
$^{13}CH_4$ addition) peaked at the oxycline (3 m) and at 7 m depth (1.0 and 0.9 μM $d^{-1}$, respectively; Fig.
1B). At 3 and 4 m depth, of all dark incubations with substrate additions (overview in Table S2), only
the addition of oxygen enhanced the methane oxidation rate (from 1.0 in the control to 1.8 μM $d^{-1}$ with
oxygen at 3 m; compare Fig. 1b and 2). Even more pronounced was the effect of light on the potential
methane oxidate rate at 3 m depth, which accelerated the methane oxidation rate to 3.9 μM $d^{-1}$ (Fig. 2).
At 4 m, the effects of light and oxygen addition were minor (0.5, 0.7 and 0.6 μM $d^{-1}$ in the control, light
and $O_2$ incubations, respectively; Fig. 2). At 5 m depth, neither light nor oxygen increased methane
oxidation rates (Fig. 2). Additions of AQDS (5, 7, 9 m), humic substances (5 and 7 m), and Fe(III) (5
and 7 m) increased the methane oxidation rate in the hypolimnion (Fig. 1). Mn(IV) and nitrite increased



the methane oxidation rate only at one specific depth (9 m and 7 m, respectively; Fig. 1). Nitrate did not
enhance methane oxidation at any of the depths (Fig. 1).

## Discussion

Despite extremely high methane concentrations in the bottom waters of Lake Lovojärvi (up to 2000
µM), the surface water methane concentration, and thus the diffusive emission potential, remained
relatively low (<0.5 µM). The pycnocline and thermocline seem to act as physical barrier, hindering
diffusive transport and containing dissolved methane in certain water layers, where the process of
methane oxidation can consume methane and diminish the methane concentration. Lake Lovojärvi
incubation experiments and the natural abundance $\delta^{13}$C-CH$_4$ signal in the water column suggest that
natural methane oxidation rates are highest at 3 and 7 m depth (Fig. 1).

### Aerobic and photosynthesis-fueled methane oxidation

Oxygen was detected down to a depth of 3.1 m (oxycline) in the surface waters of Lake Lovojärvi (Fig.
1A). Immediately below this depth, $\delta^{13}$C-CH$_4$ showed a pronounced shift to high values from -63‰ at
3.5 m to -19‰ at 3.25 m (Fig. 1B). As methanotrophs fractionate carbon isotopes (just like many other
biological reactions breaking carbon bonds), and preferentially oxidize the light carbon $^{12}$C isotopes, the
residual pool of methane becomes enriched in the heavier $^{13}$C isotopes with fractional methane turnover.
Hence, the distinct change in $\delta^{13}$C at 3-3.5 m pinpoints a hotspot of methane oxidation (Barker and Fritz,
1981). The relatively high abundance of both types of aerobic methanotrophs (i.e. gamma- and alpha-
MOB; Fig. 1D) supports the existence of a methane oxidation hotspot at the oxycline depth. Furthermore,
control methane oxidation rates were highest directly at the oxycline (~1 µM d$^{-1}$ at 3 m; Fig. 1B),
confirming that aerobic methanotrophs are most active at the oxic-anoxic transition, where both
substrates (CH$_4$ and O$_2$) overlap and conditions are most favorable for aerobic methane oxidation (Rudd
et al., 1976, Blumenberg et al., 2007; Fenchel and Blackburn, 1979). These findings correspond well
with previous studies in shallow stratified lakes, where highest methane turnover was also shown to
occur in the vicinity of the oxycline (Blees et al., 2014; Mayr et al., 2020; Milucka et al., 2015; Oswald
et al., 2015; Panganiban et al., 1979; Sundh et al., 2005).
The oxygen availability at 3 m depth is likely rate-limiting for methane oxidation, given the in situ
concentration of ± 0.5 µM (Fig. 1; Fig. S1) and the enhanced methane oxidation rate upon the addition
of oxygen (Fig. 2). Oxygen availability below the oxycline of stratified lakes is often limited due to the
low speed of diffusive oxygen transport across the oxycline (Kreling et al., 2014). In shallow Lake
Lovojärvi, another source of oxygen besides diffusive supply is likely enhancing oxygen availability to
methanotrophs, stimulating methane removal rates. A strong peak in chlorophyll a concentration was
observed at 3-4 m depth, where the light intensity was 0.3-1.14 µE m$^{-2}$ s$^{-1}$ (Fig. 2; Fig. S1), still exceeding
the threshold for photosynthesis (0.09 µE m$^{-2}$ s$^{-1}$, Gibson, 1985). At that same depth, a small peak in the



$O_2$ concentration is observed (Fig. S1A), indicating in situ oxygen production. Milucka et al. (2015) and
Oswald et al. (2015, 2016b) showed that, pending light availability, photosynthetic oxygen production
can fuel aerobic methane oxidation deep within the anoxic water column, where methane is often replete.
Produced oxygen is immediately consumed by the oxygen-limited aerobic methanotrophs, keeping the
dissolved oxygen concentrations in the water column low. Our experimental results indicate that
photosynthetically fueled methane oxidation is also a key process in methane removal in the water
column of this humic, turbid lake. The photosynthesis effect on methanotrophy is most pronounced at 3
m depth, where the methane oxidation rates increased significantly from $0.99\pm0.06$ $\mu M$ $d^{-1}$ under dark
conditions to $3.9\pm0.06$ $\mu M$ $d^{-1}$ under light conditions. Why light stimulates the methane oxidation rate
at 3 m much stronger than the addition of $O_2$ directly ($1.8\pm0.2$ $\mu M$ $d^{-1}$) remains unclear. Perhaps the
oxygen availability and consumption are better balanced in the case of light stimulation, with a direct
linkage between the production by phytoplankton and the consumption by methanotrophs, possibly even
via a physical interaction, allowing the produced $O_2$ to be more efficiently, and exclusively, used for
methane oxidation. In the case of an $O_2$ pulse, as in the oxygen addition experiment, part of the $O_2$ may
be used for non-methane-oxidation related processes (including e.g. dark respiration by phototrophs). It
is also possible that the methanotrophs were partly inhibited by the higher $O_2$ concentrations, as
methanotrophs have been suggested to be microaerophiles (Van Bodegom et al., 2001; Rudd and
Hamilton, 1975; Thottathil et al., 2019).
In incubations with water from 4 m depth, there was only a minor observable effect of $O_2$ addition and
light on the methane oxidation rate (0.5, 0.7 and 0.6 $\mu M$ $d^{-1}$ for control, light and $O_2$, respectively; Fig.
2). Oxygen availability may not be the rate-limiting factor here. The dark incubation experiments
indicate that natural methane oxidation rates are lower at 4 m than at 3 m, perhaps attributable to the
smaller methanotrophic community (Fig. 3). The addition of nitrate, nitrite and AQDS did not enhance
methane oxidation at 4 m either (Fig. 1). Hence, what the dominant terminal electron acceptor(s)
involved in methane oxidation at 4 m depth is/are, and why oxidation rates and methanotroph abundance
were lower at 4 m than at 3 m, despite the elevated methane concentrations, remains uncertain.
**Water column methane production**
The major part of methane production in Lake Lovojärvi takes place in the sediment, where high
amounts of the methane diffuse up into the water column (~2 mM at 17 m; Fig. 1B). The carbon isotopic
signature ($\delta^{13}C$ of -66‰, Fig. 1B) is indicative of a biogenic origin, the production by methanogens
(Whiticar, 1999). The concentration declines rapidly by an order of magnitude (~200 $\mu M$ at 12 m)
upwards through the pycnocline (Fig. 1B), further decreases from 12 to 6 m depth, but then shows
another maximum at 3-5 m depth. The observed peak in the methane concentration at this depth, just
below the oxycline, suggests in situ methane production (Fig. 1B). Methane is generally produced by
methanogens, anaerobic archaea that do not tolerate oxygen (Kiener and Leisinger, 1983). It would





therefore be remarkable that a zone of methane production is observed just below the oxycline, where
traces of oxygen are still present, and where oxygen is likely produced by the highly abundant
phototrophs (Fig. 2). These phototrophs may, however, not only play a role in enabling aerobic
methanotrophy, but also in methane production. Recent research has suggested that cyanobacteria are
capable of forming methane as a by-product of photosynthesis (Bižić et al., 2020), and that this might
contribute to methane emissions from oxic waters (Günthel et al., 2020). As the zone of methane
production in Lake Lovojärvi coincides with the chlorophyll peak (Fig. 1 and 2), phytoplankton-
mediated methane production may be responsible for the observed methane production near the oxycline.
Methane production under oxic conditions is, however, still highly debated. Another reasonable
explanation for the observed methane peak could be lateral transport of methane produced in sediments
in the littoral zone (Peeters et al., 2019). Archaeal methanogens of the genus *Methanoregula* were
detected in the water column, but only at 9, 11 and 17 m depth (0.1, 0.1 and 0.3 %).
**Methane oxidation in the anoxic water column**
Besides the peak in methane oxidation at 3 m depth, high methane oxidation rates were also detected at
7 m, within the anoxic part of the water column (Fig. 1). Both the incubation experiments and the $\delta^{13}$C-
CH$_4$ profile, which showed a slight increase in the $\delta^{13}$C-CH$_4$ values, suggest active methane oxidation
within the anoxic hypolimnion (4 – 9 m). The $\delta^{13}$C-CH$_4$ and methanotroph-abundance profiles also
suggest a zone of active methane oxidation between 11 and 13 m depth (Fig. 1; 3). Earlier studies have
demonstrated high methane oxidation rates in the anoxic water column of lakes, which exceeded oxic
methane oxidation rates in some cases (Blees et al., 2014; van Grinsven et al., 2020b). In the anoxic
water column of Lake Lovojärvi, nitrate, nitrite, sulfate, Fe(III) and organic matter are all present, in
varying concentrations with water column depth (Fig. 1; Fig. S2). These compounds have all been
recognized as electron acceptors potentially involved in lacustrine methane oxidation (Ettwig et al., 2010;
Kits et al., 2015a; Saxton et al., 2016; Schubert et al., 2011). Lake Lovojärvi incubation experiments
showed that nitrite, AQDS, humic substances and Fe(III) all enhanced methane oxidation at 7 m (Fig.
1). Although each of these substances may have stimulated methane oxidation directly, as terminal
electron acceptor for methane oxidation, they may also have stimulated the internal cycling of other
redox components instead, fostering methane oxidation indirectly. For example, Su et al. (2020) showed
Mn and Fe oxides can support sulfate-dependent AOM. The stimulating effect of nitrite on the methane
oxidation rate was the strongest among all substrates tested (1.5±0.1 µM d$^{-1}$ with nitrate, 0.9±0.1 µM d$^{-}$
$^{1}$ in the control experiment; Fig. 1). As methane oxidation coupled to the reduction of nitrite yields the
largest Gibbs free energy ($\Delta$G° = -1007 kJ mol$^{-1}$ CH$_4$), this form of methane oxidation may outcompete
methane oxidation coupled to the reduction of Fe(III) ($\Delta$G° = -571 kJ mol$^{-1}$ CH$_4$) or AQDS ($\Delta$G° = -41
kJ mol$^{-1}$ CH$_4$, Reed et al. 2017). Nitrite was present in the water column of Lake Lovojärvi at relatively
high concentrations (3 µM) at 7 m and below 12m (Fig. 1C), supporting the hypothesis that nitrite could
serve as an electron acceptor involved in natural methane oxidation in the Lake Lovojärvi water column.



Nitrite has been found to support methane oxidation by Candidatus *Methylomirabilis oxyfera* and
*Methylomicrobium album* (Ettwig et al., 2010; Kits et al., 2015b), but is also known to inhibit methane
oxidation at higher concentrations (Dunfield and Knowles, 1995; Hütsch, 1998). Surprisingly, nitrite
stimulated methane oxidation at 7 m but seemed to inhibit methane oxidation at all other depths (Fig.
1C). As the same amounts of nitrite were added at all depths, it is unclear why an inhibitory effect would
occur at all depths but 7 m. It may be reasonably to assume that the overall microbial community is
involved in the (de)toxification of compounds inhibitory for methanotrophs, or that the differential
response is caused by the presence of diverse methanotrophic communities, with different tolerance
levels. The methanotrophic community composition is, however, similar at 7 m compared to the other
depths (Fig. 3).
Organic material is present throughout the water column of Lake Lovojärvi (Fig. 1F). Potential
involvement of organic molecules in methane oxidation is generally tested with the humic acids
analogue AQDS (Saxton et al., 2016; Scheller et al., 2016) or a standard mixture of humic substances
provided by commercial companies or the International Humic Substances Society (van Grinsven et al.,
2020a; Valenzuela et al., 2019). In this study, both AQDS and leonardite humic acids were used as
potential electron acceptors in the incubation experiments (Fig. 1F). A difference in the effect of these
two humic substrates was observed, with the humic substances providing a stronger stimulating effect
on the methane oxidation rates than the AQDS at both 5 and 7 m (Fig. 1F). As organic matter in natural
systems is highly diverse and complex in composition, it is difficult to assess how similar the added
material is to the natural organic material present in the water column, and what causes the observed
difference between the two organic materials used in this study. Independent of the exact
mechanisms/controls with regards to the role of humics in methane oxidation, our results show, however,
that a whole spectrum of organic substrates maybe able to support AOM.

## Methane oxidizing community

Both alpha- and gammaproteobacterial methane oxidizing bacteria are present throughout the water
column according to our cell-count data (Fig. 1D). Although concentrations of methane were very low
above the oxycline (~300 nM), alpha-MOB still make up several percent of microbial community here
(3.5% of DAPI counts at 2 m). Possibly, methane reaches the upper water column via ebullition.
Methane is a gas with a low solubility and can therefore form bubbles at high sedimentary concentrations,
which are then released into the water column at instability events (Joyce and Jewell, 2003). These
bubbles exchange gas with the water during their travel upwards through the water column (Delsontro
et al., 2010). Possibly, pulses of methane are regularly delivered to the surface water via ebullition,
feeding the epilimnetic methanotrophic community. Alpha-MOB are known to predominantly occur at
higher $O_2$ levels, whereas gamma-MOB tend to prefer high $CH_4$ levels (Amaral and Knowles, 1995;
Crevecoeur et al., 2017). This zonation is visible in the Lake Lovojärvi water column, with alpha-MOB





abundance peaking at 2 m (6.8·10$^4$ cells mL$^{-1}$, Fig. 1D). The gamma-MOB abundance peaks just below
the oxycline (8.0·10$^4$ cells mL$^{-1}$, Fig. 1D), at the same depth where the peaks in $\delta^{13}$C-CH$_4$ and methane
oxidation rate were observed. A second peak in gamma-MOB abundance was observed in the deep water
column, at 13 m (13·10$^4$ cells mL$^{-1}$, Fig. 1D). These patterns are in line with a recent 16S rRNA gene
and metagenomic sequencing study in Lake Lovojärvi (Rissanen et al., 2020). Our 16S rRNA gene
sequencing data suggests that *Methylobacter* sp. represent the dominant methanotrophs in the water
column (Fig. 3), both at the oxycline and in the deep water column. This is in line with previous findings,
suggesting that *Methylobacter* sp. is a versatile methanotroph that can use both oxygen and other
substrates, such as nitrate and nitrite, for methane oxidation (van Grinsven et al., 2020b; Martinez-Cruz
et al., 2017; Smith et al., 2018). Methanotrophs belonging to the genus *Methyloparacoccus* dominate
the oxic epilimnion, but they are absent in the zone with the highest chlorophyll a concentrations (3 – 4
m; Fig. 3). Bacteria of the family Methylophilaceae were also found throughout the water column, with
the highest abundances at depths were methane oxidation occurred (Fig. 1; Fig. 3). Methylophilaceae
are methylotrophs that do not possess genes encoding for methane monooxygenases (pMMO nor
sMMO), and are therefore incapable of methanotrophy. They are known to oxidize methanol and
methylamine (Jenkins et al., 1987), which can be released by methanotrophs (Oshkin et al., 2014;
Tavormina et al., 2017; Wei et al., 2016). These may be consumed by methylotrophs belonging to the
Methylophilaceae (van Grinsven et al., 2020c), explaining the spatial co-occurrence of the two groups
in the lake water column. *Candidatus* Methylomirabilis sp. were only detected at 13 m depth, but at a
relatively large abundance (2.3 % of 16S rRNA reads).
Similar methane oxidation rates were measured at 3 and 7 m depth (1.0±0.1 and 0.9±0.1 µM d$^{-1}$,
respectively; Fig. 1B), despite a large difference in methanotroph abundance (8.5 and 2.6·10$^4$ cells mL$^{-1}$,
respectively; Fig. 1D). Water column methane oxidation rates therefore seem not necessarily coupled
to methanotroph cell numbers, but rather to cell activity rates instead.
**Conclusions**
Lake Lovojärvi is a productive humic lake. Despite the extremely high methane concentrations in its
bottom waters, it is likely not a major source of methane to the atmosphere due to effective methane
consumption in the water column. Nitrite seems to serve as the main TEA for methane oxidation at the
most active anoxic methane oxidation hotspot, yet a number of other potential organic and inorganic
electron acceptors for methane oxidation are present in the water column and were demonstrated to
stimulate AOM, demonstrating the high versatility of aerobic and anaerobic methanotrophic
communities in freshwater environments. Near the oxycline, aerobic methanotrophy is supported by
oxygen, via diffusion from above and by local production by phototrophs, and by a local input of
methane, either provided by in situ production of methane by the phototrophic community or by lateral
transport. Overall, our study in Lake Lovojärvi shows that even in shallow lakes, water column methane



oxidation can form an efficient two-step (anaerobic/aerobic) biological methane filter against methane
emissions from highly productive systems.
**Author contributions**
SG and KO wrote the original draft. SG adapted successive versions of the manuscript that led to the
final version. KO, CJ and CS were involved in designing the study, sampling campaign and experimental
setups while CS and BW developed the overall project. KO and CJ conducted the field sampling and
experiments as well as the subsequent laboratory analyses. Amplicon sequence analyses were done by
SG and JZ. CS, BW, MFL, and JZ reviewed and commented on the manuscript.
The authors declare that they have no conflict of interest.
**Acknowledgements**
The authors thank Christian Dinkel for his help in conducting the sampling campaign and operating
measuring equipment in the field. We kindly thank the staff at the Lammi Biological Station in Finland
for helping us arrange our stay there, as well as organizing a boat for the sampling campaign and the use
of the laboratory. We appreciate the support of Andreas Brand in analyzing the oxygen measurements.
We thank Patrick Kathriner, Serge Robert, David Kistler and Irene Brunner for their assistance in the
laboratory. The Swiss National Science Foundation (SNF grant 153091) and Eawag funded this work.

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
