# Peer review of "Methane oxidation in the waters of a humic-rich boreal lake stimulated by photosynthesis, nitrite, Fe(III) and humics"

_Biogeosciences, 2021_

## Referee Comment (RC2)

The manuscript by van Grinsven et al "*Methane oxidation in the waters of a humics-rich boreal lake stimulated by photosynthesis, nitrite, Fe (III) and humics*" presents a detailed study on the magnitude and control of methane ($CH_4$) oxidation in a small humic lake in Southern Finland. This study revisits some of the previously shown mechanisms regulating $CH_4$ oxidation in lakes –light, oxygen, nitrite, Fe (III), and humic substance. Although all these factors are known to stimulate $CH_4$ oxidation in freshwater lakes, assessing their roles and extent in a single lake water column is rarely attempted. Moreover, authors combined multiple tools – physico-chemical profiling, stable carbon isotopic ($\delta^{13}$C-$CH_4$) signature, near-ambient incubations with and without the addition of the above stimulants of $CH_4$ oxidation, and molecular assessment on the dynamics of $CH_4$ oxidizing bacteria (MOB). This article is well-written, and methods are adequately described. However, I feel that Introduction needs little more clarity. I suggest to re-write the introduction in such a way that objectives of the study are stated clearly and contextualized better. Moreover, I am missing a thorough discussion on the effect various factors on $CH_4$ oxidation; for example, light is stimulating $CH_4$ oxidation in this brown water lake – which is opposite to our general understanding that $CH_4$ oxidation is inhibited by light (Murase and Sugimoto, 2005, Dumestre et al 1999, Shelly et al 2017). I would suggest authors to build the discussion based on these previous studies. Probably, light play differently across depths within a lake and across the lakes based on the extent of "browning" (increase of DOC in aquatic systems) and if so, what could be the effect of ongoing "browning" on future $CH_4$ oxidation?

**Specific comments**

L7: I do not see any discernible $CH_4$ oxidation "hotspot" in the anoxic water column based on the $^{13}$C-$CH_4$ profile, although evident at ~3.0 m (Figure 2B).

L53: …and classical MO – methane oxidation (MO)?

L69-70: Please rephrase

L71-73: I feel that the problem is not very well-defined.

L79-83: I would suggest rephrasing of this entire sentence to bring clarity.

L82: TEAa

L127: Please give little more details on the headspace equilibration; for example, how did you transfer the headspace gas into vials (?). Did you add water into the bottle to replace and take out the 20 ml headspace? Please mention how exactly you did it.

L137-142: I do not see any results on the fraction oxidized ($f$) or fractionation factor ($\alpha$). Either you may provide the results or remove it from the Methods.

L206: Please provide the reasons for selecting these depths – or why you did not consider surface layers for oxidation measurements?

L230: Please provide the detection limit of the LI-CORE in the methods – I doubt how much we can rely on $0.01\mu E\ m^{-2}\ s^{-1}$.

L262: How do you define epilimnion – if it is the well-mixed surface zone, it surprising to see such a large variability in $\delta^{13}C\text{-}CH_4$ (-50‰ to -35‰) within the epilimnion. Or do you think the first value (1m?) is erroneous? Please check.

L311: "Potential Methane Oxidation Rates" instead of "Methane oxidation rates incubations"?. If you are providing mass-balance based estimation of oxidation (based on $\delta^{13}C\text{-}CH_4$), it makes sense to have "Methane oxidation rates – incubations" followed by "Methane oxidation rates – isotopic mass balance"

L333: Sentence reads strange to me – "....down to a depth of 3.1 m (oxycline) in the surface.."

L341: "control methane oxidation rates were…" rephrase the sentence.

L345: "…shallow stratified lakes,…" Please be careful with the cited references – Lake studied by Blees et al 2014 (Lake Lugano) is 288 m deep, not a shallow lake. Similarly, check other references too.

L357: "pending light availability"?

L363: 0.99 ± 0.06 – space, similarly L365.

L364-365: It seems to me that $O_2$ is consumed within a short period of time, far before the termination of incubation around 48 hrs since the initial $O_2$ concentration is only $15\mu M$ (Table S2), while light incubations continue to provide $O_2$ through photosynthesis throughout the incubation. Please look into the time course of $^{13}C\text{-}DIC$ and see whether the pattern is linear or not. Please consider this aspect.

L371: I am not convinced – Table S2 suggest $CH_4$ concentration is 15 µM, which seems to be the "optimal concentration" for highest methanotrophic activity (*see*, Thottathil et al 2019), not at a level to induce $O_2$ inhibition.

L377-378: "..perhaps attributable to the smaller methanotrophic community" – this is in contrast to L484-485 where you states that "water column methane oxidation rates therefore seems not necessarily coupled to methanotroph cell number, but rather to cell activity rates instead"

L388-389: The local peak of $CH_4$ cannot be attributed to aerobic $CH_4$ production – for example, s*ee* Donis et al 2018 which showed that such metalimnetic peaks can only be a "physically driven accumulation".

L399-401: You are suggesting two contradictory (and much debated) processes to explain the same phenomena of sub-surface $CH_4$ peaks. Based on the data from L. Lovojärvi and other similar systems, what is the most probable explanation?

L432: "It may be reasonably to assume…" requires rephrasing?

L454-459: I am not quite understanding why do pulses of $CH_4$ (that too bubble fluxes!) require to support alpha-MOB in the surface layers. In fact, recent studies have shown that Alpha-MOB are known to be regulated by oxygen concentration, particularly Alpha MOB shows high abundance at high $O_2$ concentration of ~200µM (*see* Reis et al 2019). Please look into those possibilities. Also, I doubt whether ebullition occurs at depth of 17.5 m (sampling location) to support the hypothesized bubble pulses. If bubbles are rising from 17.5 m depth, why does bubble dissolution in the water column support only the MOB at the surface layers?.

**References**

Shelley, F., Ings, N., Hildrew, A. G., Trimmer, M., & Grey, J. (2017). Bringing methanotrophy in rivers out of the shadows. Limnology and Oceanography, 62(6), 2345–2359.

Murase, J., & Sugimoto, A. (2005). Inhibitory effect of light on methane oxidation in the pelagic water column of a mesotrophic lake (Lake Biwa, Japan). Limnology and Oceanography, 50(4), 1339–1343

Dumestre, J. F., Guézennec, J., Galy-Lacaux, C., Delmas, R., Richard, S., & Labroue, L. (1999). Influence of light intensity on methanotrophic bacterial activity in Petit Saut Reservoir, French Guiana. Applied and Environmental Microbiology, 65(2), 534–539

Donis, D., Stöckli, A., Flury, S., Spangenberg, J. E., Vachon, D., & McGinnis, D. F. (2017). Full-scale evaluation of methane production under oxic conditions in a mesotrophic lake. Nature Communications, 8(1), 1661.

Reis, P. C., Thottathil, S. D., Ruiz-González, C., & Prairie, Y. T. (2020). Niche separation within aerobic methanotrophic bacteria across lakes and its link to methane oxidation rates. Environmental microbiology, 22(2), 738-751.

---

## Author Comment (AC1)

Reply by van Grinsven et al. to anonymous Referee #1

Referee comment on "Methane oxidation in the waters of a humics-rich boreal lake stimulated by photosynthesis, nitrite, Fe(III) and humics" by Sigrid van Grinsven et al., Biogeosciences Discuss., https://doi.org/10.5194/bg-2021-3-RC1, 2021

This interesting study investigates the biogeochemical methane cycle in a relatively shallow, eutrophic boreal lake using a wide range of chemical, microbiological and molecular techniques. The authors show that the lake Lovojärvi has a very active methane cycle mostly driven by upwards diffusing methane from the sediment (produced by methanogenesis) but also provide evidence for an additional source of methane within the water column near the Chla maxium. Methane seems to be efficiently consumed by the microbial community, in particular at the oxic-anoxic interface as well as in the anoxic hypolimnion. Using 13C-methane incubations, the authors show that methane oxidation in the anoxic hypolimnion seems to be coupled to in situ production of oxygen at shallower depths, while some of the tested electron acceptors appear to stimulate methane oxidation in the dark anoxic hypolimnion. Furthermore, FISH and 16S rRNA amplicon analyses indicate that alpha- and gammaproteobacterial methanotrophs appear to be the dominant methanotrophs.

Overall I find this study very interesting. Considering that boreal lakes are quite poorly characterized in respect to methane cycling, I believe that this study is a valuable addition to the current literature. The manuscript nicely highlights that light-driven methane oxidation as well as AOM coupled to other electron acceptors can be important processes in the anoxic hypolimnion of shallow boreal lakes. The evidence for photosynthesis-fueled methanotrophy appears robust und the authors do a good job discussing some of the observed anomalies (e.g. +O2 vs. light, 3 m vs. 4 m). However, I'm more skeptical about proposed stimulation of MOR by some of the amended substrates, in particular AQDS, and I feel that the authors should be more careful not to overstate the results of their incubation experiments (see point #1). Other than that, I have only minor suggestions. The manuscript is generally well written and understandable, and the Methods are rather brief but for the most part adequately described. The introduction could be more focused on methane cycling in boreal lakes in general (see point #2) and it would be helpful if the authors could provide some context around why Lovojärvi was studied (see point #3).

Re: Thank you for your kind words! We have added replies below to the separate comments.

Specific points:

**1 Stimulation of MOR in incubations**

For some incubations, there is a clear increase in MOR (e.g. light) and the data looks robust to me. However, for other incubations the stimulation is much less pronounced (e.g. Fe, humic acid) or even so small that the difference is in my opinion within the margin of error (for AQDS). Without independent biological replicate incubations, which I don't think the authors did (please correct me if I'm wrong), I am not entirely convinced that the presented data for AQDS (and possibly Fe and humic acid) conclusively show a stimulation of MOR. As it is an interesting and important conclusion, I encourage the authors to provide some additional data (e.g. statistical tests) to support their claims.

Re: To provide the reader with data that are easier to compare, also statistically, we created a barplot including error bars (Fig. 4). We agree that in the first version of the manuscript, it was difficult to determine, whether the differences between the treatments were significant. We believe this issue is solved by the addition of the new figure. We have removed the statement that AQDS stimulated methane oxidation at 9 m, because the differences between the treatments are indeed small and the error relatively large. We also checked the mentioning of a stimulating effect of AQDS on MOR in the text, to make sure the conclusions on this part were not too strongly phrased. We however believe that it is mentioned in a correct way, without speculations on the importance of this electron acceptor, and without overemphasizing, or overly stating, the effect of AQDS on methane oxidation.

**2 Introduction could be more focused on boreal lakes**

In its current form, the introduction is very general. While I agree that boreal lakes are not excessively studied, I believe that more can be said in the introduction than that "studies [...] are relatively scant". I encourage the authors to expand their introduction with more information about biological methane oxidation in boreal lakes (e.g. what is known about humic substances and why are they important, availability of other TEA, are they often Fe- and Mn-rich?).

Re: We agree that the introduction itself was relatively scant. We have now improved it strongly: we took out part of the general description of methane oxidation, especially the parts that were not relevant to the paper. We also added more information of other publications on boreal/northern lakes, including recent papers. We believe the introduction is now more targeted towards the use of different TEAs by methanotrophs on the one hand, and boreal lakes on the other hand.

**3 Boreal lakes and Lovojärvi**

Lovojärvi strikes me as a quite unique lake (presence of halocline, extreme CH4 concentration above the sediment, meromixis). Is this a typical boreal lake with typical physico-chemical features? Since the authors use their findings to make general conclusions regarding the biological methane filter and the emission potential in boreal lakes (e.g. lines 14-18), it would be important to include some discussion/description on how representative Lovojärvi is for boreal lakes in general.

Re: We have searched through literature to compare these characteristics to other boreal lakes. The number of boreal lakes investigated is relatively low, and to what we found, they differ regarding quite a large number of different characteristics: stratification, Fe-content, water color etc. It is therefore difficult to set the characteristics of a 'typical boreal lake' and also to assess, whether specific characteristics have a stronger effect on the methane oxidation process than others. We, however, consider none of the 'special' characteristics of Lovojärvi (halocline, high [CH4] in lowest layer, meromixic) to be of major or special interest for the observed methane oxidation results, also because the incubation experiments of this study were performed with water from more 'average' water layers. We therefore consider the effect of the bottom layer relatively small. Another recent study on this lake (10.1093/femsec/fiaa252) has also not classified Lovojärvi as an outlier between other (boreal) lake systems.

Minor points:

Line 64: "aerobic MOB" sounds counterintuitive in this context. Please rephrase.

Re: We have rephrased the sentence

Line 73: There is definitely more literature available on methane oxidation in boreal lakes (e.g. Rissanen et al. 2017, https://doi.org/10.1093/femsec/fix078)

Re: We have now added more references and also discussed those in more detail.

Line 164: How much water was typically filtered? Re: 15 mL, this is added to the text now.

Line 174: How many cells were counted? Re: Added to the text now: *(260 - 550 cells counted per sample, distributed over 20 randomly chosen fields of view).*

Line 191: Include some information regarding sequencing depth (either here or as a table)

Re: Below a table with the sequencing depth. This table could be added to the manuscript as supplemental table, but we don't consider this directly necessary.

| Depth | Reads |
|-------|-------|
| 2m | 26810 |
| 2.5m | 29076 |
| 3m | 30087 |
| 3.5m | 33399 |
| 4m | 32198 |
| 5m | 27523 |
| 6m | 30009 |
| 7m | 32379 |
| 11m | 123313 |
| 13m | 35670 |
| 17m | 23820 |

Line 233: The NOx profiles are quite stunning. I assume that the nitrate and nitrite peak close to the base of the oxycline are due to microbial ammonia oxidation. But what could be the source of nitrite in the bottom water?

Re: We have not included information about the N-profiles in the manuscript because it is not the scope of the paper. The nitrate profile is indeed consistent with nitrate regeneration through ammonium oxidation (producing the peak), as well as upward diffusion into the photic zone and downward diffusion into the denitrification zone, where it is being consumed in both locations. Nitrite is expected to occur as a reaction intermediate. We have, however, not determined which NOx-related processes occur in the water column as this was not within the scope of our research.

lines 292: The meaning of 'other Methylococcaceae' is unclear to me. Please specify.

Re: We have now clarified this in the revised text: "*At 3.5, 13 and 17 m, respectively 0.3, 0.1 and 0.3 % of 'other Methylococcaceae', specified as 16S rRNA sequence assigned to the family Methylococcaceae but not to the above-mentioned genera, were found.*"

lines 292-296: I'm confused. Methylocystaceae abundance seems low but this sentence suggests to me that they might be high since you detected unknown Rhizobiales bacteria? Please clarify.

Re: We have rephrased here to clarify: "*30 – 35 % of the Alphaproteobacterial reads at 2 – 3 m depth were, however, assigned to unknown bacteria of the Rhizobiales order, the order to which the alpha-MOB belong (Fig. S4). Possibly, part of these unknown Rhizobiales-assigned sequences belongs to methane oxidizing bacteria.*"

lines 283-301: It's not clear to me according to what logic the abundances of different methanotrophs, methylotrophs and some seemingly random taxa (Acidoferax, Planctomycetaceae, Rhizobiales) are listed one after the other. Please restructure. Also, some of these groups are never discussed and it's not clear why they are specifically mentioned here.

Re: We have listed the abundances of gamma-MOB and alpha-MOB, one after an other here, following the same order as in the previous paragraph on the FISH-results. Furthermore, we have added the relative abundance of other groups that we considered relevant for the reader (i.e.methylotrophs). We originally also included the other highly abundant microbial groups to provide a more complete picture of the microbial community in general, but we agree that it may be better to focus on the taxa that are most relevant in the context of this methane paper, which we do now.

Line 290: Were you able to observe any filamentous gamma-MOB using FISH?

Re: We did not find any evidence of filamentous gamma-MOB in the samples analyzed with the specified CARD-FISH probes. Both hybridized gamma- and alpha MOB were circular in shape with an approximate average cell size of 2 μm and 1 μm, respectively.

line 311: "natural conditions" suggests that different light intensities were used for incubations from 3m and 7, please clarify.

Re: We agree this was unclear, and we have now adapted it to "control", which is the same naming as used in the graphs. "*Methane oxidation under "control" conditions (dark, starting concentration ~50 μM $CH_4$ after $^{13}CH_4$ addition) peaked at the oxycline (3 m) and at 7 m depth (1.0 and 0.9   M $d^{-1}$, respectively; Fig. 1B).*"

Line 319: Given the uncertainties in Table S3, AQDS 5m MOR increase does not look significant.

Re: We have now added Fig. 4, which allows for an easier comparison between the different treatments compared to the previous Fig. 1. Now, it is quite obvious that at 5 and 7 m, the uncertainties are relatively small. At 4 m and 9 m, there is indeed not enough certainty to make a robust statement on the effect of AQDS, so we have removed it from the text. We have also taken out 'in the hypolimnion', as 5 and 7 m do not cover the whole hypolimnion. "*Additions of AQDS, humic substances, and Fe(III) increased the methane oxidation rate at 5 and 7 m depth (Fig. 1).*"

line 348: What is meant by a concentration of +/- 0.5 uM ?

Re: We used this +/- because the O2 concentration gradient is very steep around 3m depth. But we recognize it is unclear and as the concentration is 0.5 uM at 3m, we have now removed the +/-.

line 392-401: It would be interesting if the authors could slightly expand on methanogenesis by phototrophs by including some brief speculation what cyanobacterial groups could be responsible for this (using the amplicon data).

Re: We are not aware of any studies that show that certain groups of cyanobacteria can produce methane, and others cannot. We therefore refrain from adding more speculation than needed to the manuscript.

Lines 486-488: The contribution of methanotrophs is indeed important, however, I suppose the halocline also plays an important role?

Re: We have adapted the text to include this: "*, it is likely not a major source of methane to the atmosphere due to effective methane consumption in the water column, combined with limited gas diffusion from the deep water layers.*"

Fig 1: This is quite a busy figure that could use some improvement. I suggest that change the scale of the x-axis for oxygen to highlight the O2 dynamics the lower concentration range (as shown in Fig. S1). In panel A, it looks as if oxygen concentration increases slightly in the hypolimnion, please comment (also in Fig. 2). In panel C, value for MOR – NO2 at 7 m is clearly <1.5 while table S3 shows a value of 1.54. Please explain error bars in legend.

Overall I suggest that the authors revise it to improve clarity. For example: i) not all x- axis same length (panesl C and D) or ii) error bars sometimes not visible.

Re: We agree that this figure was not clear enough and the information was hard to interpret. We have therefore taken the MOR-data out of this figure, and now show them in a separate barplot (Fig. 4). We have decided to leave the full oxygen profile in Fig. 1, but we have moved the zoomed-in plot of Fig. S1 to the main text. It is now combined within Fig. 2.

We have also looked into the O2 concentration data in more detail. We used two different oxygen sensors (described in the method section), and in the original figure the data from the higher-level sensor was used, also for the hypolimnion values. This sensor is, however, not suitable for the low concentrations below the oxycline. We have now adapted the figures in the manuscript to show the correct datasets, and have also clarified this in the figure's caption.

Fig 2: In my opinion, the y-axis could be limited to 10 m in order to focus on the upper water column.

RE: We agree and have adapted Fig. 2. Now, the details of the upper water column are better visible. Thank you for mentioning.

Fig 3. Only cosmetic, but there is an offset between lines and symbols.

Re: Adapted

---

## Author Comment (AC2)

Reply of van Grinsven et al. to Anonymous reviewer 2

The manuscript by van Grinsven et al "Methane oxidation in the waters of a humics-rich boreal lake stimulated by photosynthesis, nitrite, Fe (III) and humics" presents a detailed study on the magnitude and control of methane (CH4) oxidation in a small humic lake in Southern Finland. This study revisits some of the previously shown mechanisms regulating CH4 oxidation in lakes –light, oxygen, nitrite, Fe (III), and humic substance. Although all these factors are known to stimulate CH4 oxidation in freshwater lakes, assessing their roles and extent in a single lake water column is rarely attempted. Moreover, authors combined multiple tools – physicochemical profiling, stable carbon isotopic ($\delta$13C-CH4) signature, near-ambient incubations with and without the addition of the above stimulants of CH4 oxidation, and molecular assessment on the dynamics of CH4 oxidizing bacteria (MOB). This article is well-written, and methods are adequately described. However, I feel that Introduction needs little more clarity. I suggest to re-write the introduction in such a way that objectives of the study are stated clearly and contextualized better. Moreover, I am missing a thorough discussion on the effect various factors on CH4 oxidation; for example, light is stimulating CH4 oxidation in this brown water lake – which is opposite to our general understanding that CH4 oxidation is inhibited by light (Murase and Sugimoto, 2005, Dumestre et al 1999, Shelly et al 2017). I would suggest authors to build the discussion based on these previous studies. Probably, light play differently across depths within a lake and across the lakes based on the extent of "browning" (increase of DOC in aquatic systems) and if so, what could be the effect of ongoing "browning" on future CH4 oxidation?

Re: We have revised the introduction. We took out part of the general description of methane oxidation, especially the parts that were not relevant to the paper. We also added more information of other publications on boreal/northern lakes, including recent papers. We believe the introduction is now more aimed towards the use of different TEAs by methanotrophs on the one hand, and boreal lakes on the other hand.

Although there are indeed papers showing light inhibition of methanotrophy, there is a larger number of papers by now that show light-stimulated methane oxidation, via the pathway of coupled photosynthesis-aerobic MO (first discovered by Milucka et al. 2015, but also shown by Oswald et al.2015, Kallistova et al. 2019, Savvichev et al. 2019). The finding of light-stimulated MO in this lake therefore fits well within the expectation based on previous publications, and is strongly supported by the observed peak in chlorophyll. The light intensity in the lake at the methane oxidation peak (3 m), is also already strongly decreased compared to the surface (Fig. 2, also stated in the text: *". A strong peak in chlorophyll a concentration was observed at 3 – 4 m depth, where the light intensity was 0.3 – 1.14 $\mu E\ m^{-2}\ s^{-1}$ "*), which may also prevent light inhibition.

To properly address the topic of browning, additional incubations would need to be performed. With the incubation experiment data in hand (only one light intensity), we have no information on the effect of diminishing light. We also don't have specific information on the oxygen demand of methane oxidation, so we can't perform a proper calculation to assess how much photosynthesis/O2 production would be required to still sustain methane oxidation, and at what rate. We therefore consider it too speculative to discuss the effect of ongoing browning on CH4 oxidation in the manuscript.

Specific comments L7:
I do not see any discernible CH4 oxidation "hotspot" in the anoxic water column based on the 13C-CH4 profile, although evident at ~3.0 m (Figure 2B).

Re: The 13C-CH4 profile shows, besides the distinct peak at 3 m, two smaller peaks at 7-8 m and 11-12 m. However, the reviewer is right in that the word 'hotspot' is perhaps too strong. We have therefore adapted the sentence: *"$\delta^{13}C$-$CH_4$ profiling of the water column revealed a methane-oxidation hotspot just below the oxycline and zones of methane oxidation within the anoxic water column."*

L53: …and classical MO – methane oxidation (MO)?
Re: adjusted, it now reads *"The hybrid metabolism of Methylomirabilis oxyfera combines partial denitrification ($NO_2^-$ to NO) and classical aerobic methane oxidation, fueled by internal $O_2$ generation"*

L69-70: Please rephrase
Re: adapted, to: *"While oxygen supplied by episodic down-welling of cold $O_2$-laden water (Blees et al., 2014), or low-light photosynthesis (Milucka et al., 2015; Oswald et al., 2015) may explain this phenomenon to some degree, methane oxidation may also be coupled to the reduction of other electron acceptors than $O_2$, such as $NO_X$ (Deutzmann et al., 2014; Graf et al., 2018; Oswald et al., 2016b), Fe(III) (Norði et al., 2013; Sivan et al., 2011), Mn(IV) (Crowe et al., 2011; Oswald et al., 2016a) and humic substances (Valenzuela et al., 2019) ."*

L71-73: I feel that the problem is not very well-defined.
Re: This part of the introduction has been completely changed in the revised version of the manuscript.

L79-83: I would suggest rephrasing of this entire sentence to bring clarity.
Re: This part of the introduction has completely changed in the revised version of the manuscript.

L82: TEAa
RE: adapted

L127: Please give little more details on the headspace equilibration; for example, how did you transfer the headspace gas into vials (?). Did you add water into the bottle to replace and take out the 20 ml headspace? Please mention how exactly you did it.

Re: We have expanded the text to include more details on the procedure. *"A headspace was created by exchanging 20 mL lake water with 20 mL $N_2$ gas. The bottles were then left for at least 24 hours to equilibrate the gas content between the gas and water phase. Afterwards, headspace gas samples were used to measure the $CH_4$ concentration by gas chromatography (GC; Agilent 6890N, Agilent Technologies) using a Carboxen 1010 column (30 m x 0.53 mm, Supelco), a flame ionization detector and an auto-sampler (Valco Instruments Co. Inc.). Resulting headspace concentrations were converted to dissolved water-phase $CH_4$ by applying calculated Bunsen solubility coefficients (Wiesenburg and Guinasso, 1979)"*

L137-142: I do not see any results on the fraction oxidized (f) or fractionation factor ($\alpha$). Either you may provide the results or remove it from the Methods.
Re: Thank you for noticing. We have removed it from the Methods.

L206: Please provide the reasons for selecting these depths – or why you did not consider surface layers for oxidation measurements?
Re: We have added the following sentence to the text, to clarify our choice for these depths: *"These depths were selected based on their expected relevance for the methane cycle: previous research has*

*repeatedly shown the highest methane oxidation rates to occur around the oxycline (Blees et al., 2014; Mayr et al., 2020; Milucka et al., 2015; Oswald et al., 2015; Panganiban et al., 1979; Sundh et al., 2005)."*

L230: Please provide the detection limit of the LI-CORE in the methods – I doubt how much we can rely on 0.01µE m-2 s -1 .

Re: We have now adapted this statement to: "Light diminished between 5 and 6.6 m (0.05 – 0.01 µE m$^{-2}$ s$^{-1}$; Fig. 2)."

L262: How do you define epilimnion – if it is the well-mixed surface zone, it surprising to see such a large variability in $\delta$13C-CH4 (-50‰ to -35‰) within the epilimnion. Or do you think the first value (1m?) is erroneous? Please check.

Re: We define the epilimnion to 3 meters, which is the depth where oxygen becomes depleted. The measured $\delta$13C-CH4 value of -50 is close to the atmospheric value of -47. Methane may also have been transported from the littoral zone, or come from below. We therefore don't think that the values are erroneous or problematic.

L311: "Potential Methane Oxidation Rates" instead of "Methane oxidation rates incubations"?. If you are providing mass-balance based estimation of oxidation (based on $\delta$ 13C-CH4), it makes sense to have "Methane oxidation rates – incubations" followed by "Methane oxidation rates – isotopic mass balance"
Re: Paragraph title adjusted

L333: Sentence reads strange to me – "....down to a depth of 3.1 m (oxycline) in the surface.."
Re: adjusted: *"Oxygen was detected down to a depth of 3.1 m (oxycline) within Lake Lovojärvi (Fig. 1A and XX2)"*

L341: "control methane oxidation rates were…" rephrase the sentence.
RE: adapted

L345: "…shallow stratified lakes,…" Please be careful with the cited references – Lake studied by Blees et al 2014 (Lake Lugano) is 288 m deep, not a shallow lake. Similarly, check other references too.
Re: Thank you for noticing. We have now removed the word "shallow", because it is not important whether the lakes are shallow; the preceding text is about the oxic-anoxic transition zone, which occurs both in deep and shallow lakes.

L357: "pending light availability"? Re: adapted

L363: 0.99 ± 0.06 – space, similarly L365. Re: adapted

L364-365: It seems to me that O2 is consumed within a short period of time, far before the termination of incubation around 48 hrs since the initial O2 concentration is only 15µM (Table S2), while light incubations continue to provide O2 through photosynthesis throughout the incubation. Please look into the time course of 13C-DIC and see whether the pattern is linear or not. Please consider this aspect.

Re: The methane oxidation rate is determined based on the linear increase in 13C-DIC, as is also described in the material and methods section ("Methane oxidation rates were estimated by linear

regression of the change of $^{13}$C-DIC over the experimental interval," ). In none of the incubations with O2 did we observe a non-linear trend in 13C-DIC production over the 48 hour incubation period. This suggests that oxygen-limitation did not occur and that substrate oxygen was not used up over the course of the experiment. The data of 13C-DIC over time is currently not shown in the manuscript, but could be included as a supplementary figure, at the discretion of the AE. We, however, think it is not needed.

L371: I am not convinced – Table S2 suggest CH4 concentration is 15 µM, which seems to be the "optimal concentration" for highest methanotrophic activity (see, Thottathil et al 2019), not at a level to induce O2 inhibition.
Re: In this paragraph, we present several possible options for the observed difference in MOR between the oxic and light incubation. We do not state that inhibition is *the* explanation. But we think that it is worth mentioning, among others, the possibility of O2 inhibition, as for several methanotrophs, their ideal oxygen conditions are not known yet.

L377-378: "..perhaps attributable to the smaller methanotrophic community" – this is in contrast to L484-485 where you states that "water column methane oxidation rates therefore seems not necessarily coupled to methanotroph cell number, but rather to cell activity rates instead"
Re: We have removed the first sentence from the discussion.

L388-389: The local peak of CH4 cannot be attributed to aerobic CH4 production – for example, see Donis et al 2018 which showed that such metalimnetic peaks can only be a "physically driven accumulation".
Re: There is a lot of ongoing debate on this topic, and both papers on physically driven accumulation and cyanobacterial production are being published ((https://doi.org/10.1002/lno.11557, https://www.nature.com/articles/s41467-021-21215-2, https://www.nature.com/articles/s41467-021-21216-1, https://www.nature.com/articles/s41598-018-36530-w), including a paper by the same author D. Donis, stating "internal methane production in well-oxygenated surface water is an important source for surface-water methane" https://www.nature.com/articles/s41467-019-13320-0. Because of the still ongoing debate, we decided to display both possibilities, of in situ production and of lateral transport (*"Another reasonable explanation for the observed methane peak could be lateral transport of methane produced in sediments in the littoral zone"*) to allow the reader to also judge on this matter, as we cannot be certain, which process is responsible for the methane profile that we observe.

L399-401: You are suggesting two contradictory (and much debated) processes to explain the same phenomena of sub-surface CH4 peaks. Based on the data from L. Lovojärvi and other similar systems, what is the most probable explanation?
Re: Based on the data in hand from Lovojärvi, we cannot define which process is leading to the observed CH4 accumulation (or whether both mechanisms are at work simultaneously). We therefore discuss both possibilities in the text. Overall, the co-occurrence between the phytoplankton peak and the methane maximum is striking, but it there not sufficient conclusive evidence to suggest in the manuscript that cyanobacterial methane production is indeed occurring and is the sole mechanism explaining the CH4 conentration peak.

L432: "It may be reasonably to assume…" requires rephrasing? Re: adjusted

L454-459: I am not quite understanding why do pulses of CH4 (that too bubble fluxes!) require to support alpha-MOB in the surface layers. In fact, recent studies have shown that AlphaMOB are known to be regulated by oxygen concentration, particularly Alpha MOB shows high abundance at high O2 concentration of ~200µM (see Reis et al 2019). Please look into those possibilities. Also, I

doubt whether ebullition occurs at depth of 17.5 m (sampling location) to support the hypothesized bubble pulses. If bubbles are rising from 17.5 m depth, why does bubble dissolution in the water column support only the MOB at the surface layers?.

Re: The reason why we specifically mention ebullition in relationship with the alpha-MOB in the surface layers, is that little to no methane seems (based on the analysis of concentration profiles) to reach the surface water layer via diffusion. At the depths where gamma-MOB were observed, diffusion seemed to be the main methane-delivering process. We do not mean to imply that bubble dissolution only occurs in the surface; if it happens, it happens everywhere, yet only in the surface waters, it is may be only way of methane delivery to the MOB. We have now adjusted the text to make this clearer: *"Possibly, these methanotrophs are supported by methane that reaches the upper water column via ebullition, in contrast to the continuous methane supply by diffusion to MOB in the lower water layers."* We have also added a sentence that includes the option that methane is delived to the alpha MOB laterally, from the littoral zone: *"Another possibility is the influx of methane from the littoral zone, via lateral transport."* The relationship between O2 and alpha-MOB was already mentioned in the text (*"Alpha-MOB are known to predominantly occur at higher $O_2$ levels, whereas gamma-MOB tend to prefer high $CH_4$ levels (Amaral and Knowles, 1995; Crevecoeur et al., 2017). This zonation is visible in the Lake Lovojärvi water column, with alpha-MOB abundance peaking at 2 m ($6.8 \cdot 10^4$ cells mL$^{-1}$, Fig. 1D).*")

References
Shelley, F., Ings, N., Hildrew, A. G., Trimmer, M., & Grey, J. (2017). Bringing methanotrophy in rivers out of the shadows. Limnology and Oceanography, 62(6), 2345–2359.
Murase, J., & Sugimoto, A. (2005). Inhibitory effect of light on methane oxidation in the pelagic water column of a mesotrophic lake (Lake Biwa, Japan). Limnology and Oceanography, 50(4), 1339–1343
Dumestre, J. F., Guézennec, J., Galy-Lacaux, C., Delmas, R., Richard, S., & Labroue, L. (1999). Influence of light intensity on methanotrophic bacterial activity in Petit Saut Reservoir, French Guiana. Applied and Environmental Microbiology, 65(2), 534–539
4 Donis, D., Stöckli, A., Flury, S., Spangenberg, J. E., Vachon, D., & McGinnis, D. F. (2017). Full-scale evaluation of methane production under oxic conditions in a mesotrophic lake. Nature Communications, 8(1), 1661. Reis, P. C., Thottathil, S. D., Ruiz-González, C., & Prairie, Y. T. (2020). Niche separation within aerobic methanotrophic bacteria across lakes and its link to methane oxidation rates. Environmental microbiology, 22(2), 738-751.

---

## Author Comment (AC3)

**Reply from van Grinsven et al. to Community Comment on bg-2021-3**

Antti Rissanen

Community comment on "Methane oxidation in the waters of a humics-rich boreal lake stimulated by photosynthesis, nitrite, Fe(III) and humics" by Sigrid van Grinsven et al., Biogeosciences Discuss., https://doi.org/10.5194/bg-2021-3-CC1, 2021

It was really interesting to read a study on methane oxidation conducted in the same study lake as our recent study (Rissanen et al. FEMS Microb Ecol, Volume 97, Issue 2, February 2021, fiaa252). While our study was focused on the genetic potential of methanotrophs, this study provides very valuable novel information on the various factors affecting the activity of methanotrophs in boreal lakes. Here are some minor suggestions, which I think could further improve the manuscript:

Re: We very much appreciate your time and effort in reading and commenting on our study. We also highly value the paper suggestions you delivered. We have replied to your specific comments below. Some of these refer to comments in reply to the reviewers, which can be viewed in separate replies in the Biogeosciences discussion forum.

Line 49-52. It is perhaps worth to mention here study by Zheng et al. 2020 on extracellular electron transfer from methane to Fe-mineral by Methylomonas in hypoxic conditions.

https://pubs.acs.org/doi/abs/10.1021/acs.estlett.0c00436

Re: Reference added.

Line 69-70. Either "can play" or "are likely to play" Re: adapted

Line 73-74 and in especially discussion. Maybe it would be relevant to mention and discuss your results also in the light of our previous study from 2018, which was conducted in the nearby humic boreal lakes, where we also studied lake water methane oxidation with amendments of various EAs (incl. NO3, metals, organic EAs) and in different light conditions, and also studied the community structure and genetic potential of methanotroph communities (Rissanen et al. 2018):

https://www.int-res.com/abstracts/ame/v81/n3/p257-276/

Furthermore, study by Kallistova et al. (2018) might be also relevant to mention and discuss. They also studied methane oxidation and MOB communities in water column of a boreal lake.

https://www.int-res.com/abstracts/ame/v82/n1/p1-18/

Re: Both papers were added to the manuscript.

Line 87-. Study site. It is maybe worth to mention the historical anthropogenic effects, the soaking of flax and hemp, which potentially have contributed to the (chemical) stratification in the lake. In the Finnish publication (Tolonen et al. 1976), it is mentioned in Finnish that "Hampun ja myöhemmin myös pellavan liotus nopeuttivat läheisen Lovojärven pilaantumista jo rautakaudella (Huttunen & Tolonen 1975)." = Soaking of hemp and later also soaking of flax accelerated the pollution of nearby Lake Lovojärvi already during Iron Age".

Tolonen K, Tolonen M, Honkasalo L et al. Esihistoriallisen ja historiallisen maankäytön vaikutuksesta Lammin Lampellonjärven kehitykseen. Luonnon Tutkija. 1976;80:1–15 (in Finnish with English abstract):

https://www.researchgate.net/publication/311665698_Esihistoriallisen_ja_historiallisen_maankayton_vaikutus_Lammin_Lampellonjarven_kehitykseen_The_influence_of_of_prehistoric_and_historic_land_use_on_Lake_Lampellonjarvi_South_Finland

Unfortunately, I could not find the original reference of Huttunen & Tolonen 1975.

Re: Thank you, for directing us to these interesting references. We have now added this information to the "Study site" section of the "Materials and methods".

Line 118. were fixed Re: corrected
Line 382- What about archaea-driven methanogenesis in anoxic micro-niches?

Re: This is indeed an interesting possibility, but as we can only speculate about it, without any supporting data regarding anoxic micro-niches, we prefer not elaborate on this in the discussion.

Also more generally, there are recent studies suggesting that also methanogenic archaea can oxidize methane anaerobically, e.g. via extracellular electron transfer to solid EAs (iron minerals, organic EAs, anode in bioelectrochemical systems). Maybe worth to mention and discuss. See, e.g.

https://www.sciencedirect.com/science/article/abs/pii/S1385894720328199
https://pubmed.ncbi.nlm.nih.gov/28965392/

Re: Although the potential involvement of methanogens in methanotrophy is really fascinating, the potential contribution of methanogens here would be very low, as they are detected only at certain depths, and at much lower relative abundances than bacterial methanotrophs ("Archaeal methanogens of the genus *Methanoregula* were detected in the water column, but only at 9, 11 and 17 m depth (0.1, 0.1 and 0.3 %)." and Fig. 3). We therefore choose not to add this aspect to the discussion.

Line 468-470. Our study in the same study lake (which has been cited but not in this context) detected genetic potential for nitrate/nitrite/NO – reduction as well as for extracellular electron transfer (to metal minerals and organic EAs) in metagenome assembled genomes of Methylococcales (incl. Methylobacter sp. and the Crenothrix-type MOB), which supports the results of this study on enhancement of methane oxidation by these various EAs. See:

https://academic.oup.com/femsec/article/97/2/fiaa252/6034011

Re: Thank you for pointing this out. We have now included these findings, which wonderfully support our results, into the discussion (*"These patterns are in line with a recent 16S rRNA gene and metagenomic sequencing study in Lake Lovojärvi (Rissanen et al., 2020), which also showed the presence of nitrite reduction genes in Methylococcales metagenome assemblies of the water column, as well as genes related to extracellular electron transfer"*)

Line 473-481. Microbial interactions. Maybe worth to include and discuss also the results by Cabrol et al. 2020. They studied anaerobic methane oxidation (AOM) and MOB communities in water

columns of northern lakes and found correlation between Methylococcales and OTUs within Methylotenera, Geothrix and Geobacter genera which indicated that AOM might occur in an interaction between MOB, denitrifiers and iron- cycling partners.

https://www.sciencedirect.com/science/article/pii/S0048969720331053

Re: There are indeed several studies showing the cooccurrence of MOB with certain species, of which especially Methylotenera is often found cooccurring with MOBs. We have, however, decided to not further elaborate on the aspect of possible microbial interactions, as we have little data to support it, only relative abundance data.

Figure 1. Is there a slight increase in O2 towards the deepest depths (from appr. 15 m to the deepest depth)? If there is, what is causing it?

Re: See above the comment on this, in reply to a comment by reviewer #1. The figures are adapted now, to contain the correct data set, of the trace-sensitive oxygen sensor.

References:
Kortelainen et al. 2000. Numbers as subscripts for $CH_4$, $CO_2$ and $N_2O$

Mutyaba 2012. Maybe it could be mentioned that it is Master of Science thesis. And perhaps provide a link to it. https://jyx.jyu.fi/handle/123456789/40735

Rissanen et al. DOI-link is missing.